# Predicting the Generalization Gap in Deep Networks with Margin Distributions

**Yiding Jiang** [*], **Dilip Krishnan, Hossein Mobahi, Samy Bengio**
Google AI
`{ydjiang, dilipkay, hmobahi, bengio}@google.com`

## Abstract

Recent research has demonstrated that deep neural networks can perfectly fit randomly labeled data, but with very poor accuracy on held out data. This phenomenon indicates that loss functions such as cross-entropy are not a reliable indicator of generalization. This leads to the crucial question of how generalization gap can be predicted from training data and network parameters. In this paper, we propose such a measure, and conduct extensive empirical studies on how well it can predict the generalization gap. Our measure is based on the concept of margin distribution, which are the distances of training points to the decision boundary. We find that it is necessary to use margin distributions at multiple layers of a deep network. On the CIFAR-10 and the CIFAR-100 datasets, our proposed measure correlates very strongly with the generalization gap. In addition, we find the following other factors to be of importance: normalizing margin values for scale independence, using characterizations of margin distribution rather than just the margin (closest distance to decision boundary), and working in log space instead of linear space (effectively using a product of margins rather than a sum). Our measure can be easily applied to feedforward deep networks with any architecture and may point towards new training loss functions that could enable better generalization.

## 1 Introduction

Generalization, the ability of a classifier to perform well on unseen examples, is a desideratum for progress towards real-world deployment of deep neural networks in domains such as autonomous cars and healthcare. Until recently, it was commonly believed that deep networks generalize well to unseen examples. This was based on empirical evidence about performance on held-out dataset. However, new research has started to question this assumption. Adversarial examples cause networks to misclassify even slightly perturbed images at very high rates (Goodfellow et al., 2014; Papernot et al., 2016). In addition, deep networks can overfit to arbitrarily corrupted data (Zhang et al., 2016), and they are sensitive to small geometric transformations (Azulay & Weiss, 2018; Engstrom et al., 2017). These results have led to the important question about how the generalization gap (difference between train and test accuracy) of a deep network can be predicted using the *training data* and *network parameters*. Since in all of the above cases, the training loss is usually very small, it is clear that existing losses such as cross-entropy cannot serve that purpose. It has also been shown (e.g. in Zhang et al. (2016)) that regularizers such as weight decay cannot solve this problem either.

Consequently, a number of recent works (Neyshabur et al., 2017b; Kawaguchi et al., 2017; Bartlett et al., 2017; Poggio et al., 2017; Arora et al., 2018) have started to address this question, proposing generalization bounds based on analyses of network complexity or noise stability properties. However, a thorough empirical assessment of these bounds in terms of how accurately they can predict the generalization gap across various practical settings is not yet available.

In this work, we propose a new quantity for predicting generalization gap of a feedforward neural network. Using the notion of margin in support vector machines (Vapnik, 1995) and extension to deep networks (Elsayed et al., 2018), we develop a measure that shows a strong correlation with generalization gap and significantly outperforms recently developed theoretical bounds on

---

[*]Work done as part of the Google AI Residency.
Data and relevant code are at https://github.com/google-research/google-research/tree/master/demogen

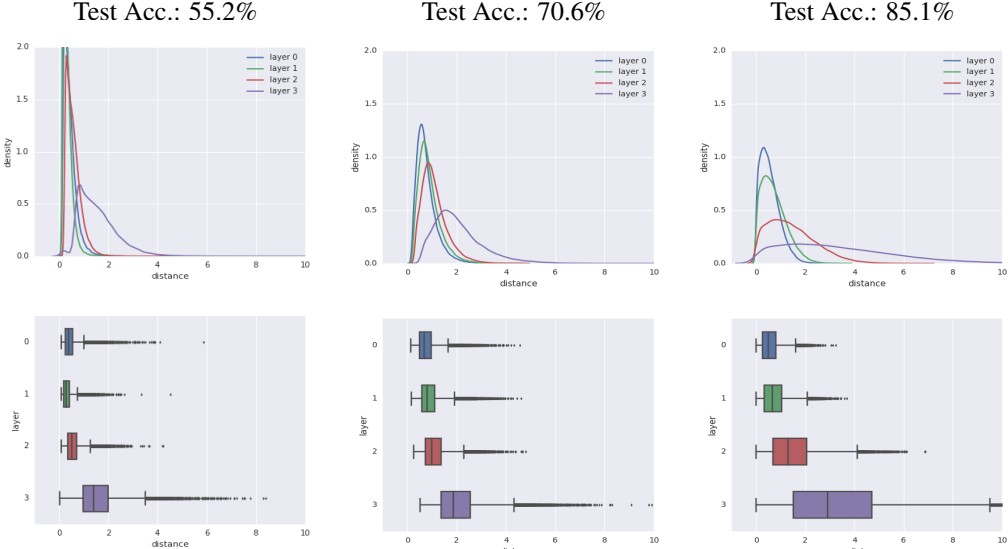

Figure 1: (Best seen as PDF) Density plots (top) and box plots (bottom) of normalized margin of three convolutional networks trained with cross-entropy loss on CIFAR-10 with varying test accuracy: left: 55.2%, middle: 70.6%, right: 85.1%. The left network was trained with 20% corrupted labels. Train accuracy of all above networks are close to 100%, and training losses close to zero. The densities and box plots are computed on the training set. Normalized margin distributions are strongly correlated with test accuracy (moving to the right as accuracy increases). This motivates our use of normalized margins at all layers. The (Tukey) box plots show the median and other order statistics (see section 3.2 for details), and motivates their use as features to summarize the distributions.

generalization[2]. This is empirically shown by studying a wide range of deep networks trained on the CIFAR-10 and CIFAR-100 datasets. The measure presented in this paper may be useful for a constructing new loss functions with better generalization. Besides improvement in the prediction of the generalization gap, our work is distinct from recently developed bounds and margin definitions in a number of ways:

1. These recently developed bounds are typically functions of weight norms (such as the spectral, Frobenius or various mixed norms). Consequently, they cannot capture variations in network topology that are not reflected in the weight norms, e.g. adding residual connections (He et al., 2016) *without careful additional engineering* based on the topology changes. Furthermore, some of the bounds require specific treatment for nonlinear activations. Our proposed measure can handle any feedforward deep network.

2. Although some of these bounds involve margin, the margin is only defined and measured at the output layer (Bartlett et al., 2017; Neyshabur et al., 2017b). For a deep network, however, margin can be defined at any layer (Elsayed et al., 2018). We show that measuring margin at a single layer does not suffice to capture generalization gap. We argue that it is crucial to use margin information across layers and show that this significantly improves generalization gap prediction.

3. The common definition of margin, as used in the recent bounds e.g. Neyshabur et al. (2017b), or as extended to deep networks, is based on the closest distance of the training points to the decision boundary. However, this notion is brittle and sensitive to outliers. In contrast, we adopt *margin distribution* (Garg et al., 2002; Langford & Shawe-Taylor, 2002; Zhang & Zhou, 2017; 2018) by looking at the entire distribution of distances. This is shown to have far better prediction power.

4. We argue that the direct extension of margin definition to deep networks (Elsayed et al., 2018), although allowing margin to be defined on all layers of the model, is unable to capture

---

[2]In fairness, the theoretical bounds we compare against were designed to be provable *upper bounds* rather than estimates with *low expected error*. Nevertheless, since recent developments on characterizing the generalization gap of deep networks are in form of upper bounds, they form a reasonable baseline.

generalization gap without proper normalization. We propose a simple normalization scheme that significantly boosts prediction accuracy.

## 2 RELATED WORK

The recent seminal work of Zhang et al. (2016) has brought into focus the question of how generalization can be measured from training data. They showed that deep networks can easily learn to fit randomly labeled data with extremely high accuracy, but with arbitrarily low generalization capability. This overfitting is not countered by deploying commonly used regularizers. The work of Bartlett et al. (2017) proposes a measure based on the ratio of two quantities: the margin distribution measured at the output layer of the network; and a spectral complexity measure related to the network's Lipschitz constant. Their normalized margin distribution provides a strong indication of the complexity of the learning task, e.g. the distribution is skewed towards the origin (lower normalized margin) for training with random labels. Neyshabur et al. (2017b;a) also develop bounds based on the product of norms of the weights across layers. Arora et al. (2018) develop bounds based on noise stability properties of networks: more stability implies better generalization. Using these criteria, they are able to derive stronger generalization bounds than previous works.

The margin distribution (specifically, boosting of margins across the training set) has been shown to correspond to generalization properties in the literature on linear models (Schapire et al., 1998): they used this connection to explain the effectiveness of boosting and bagging techniques. Reyzin & Schapire (2006) showed that it was important to control the complexity of a classifier when measuring margin, which calls for some type of normalization. In the linear case (SVM), margin is naturally defined as a function of norm of the weights Vapnik (1995). In the case of deep networks, true margin is intractable. Recent work (Elsayed et al., 2018) proposed a linearization to approximate the margin, and defined the margin at any layer of the network. (Sokolic et al., 2016) provide another approximation to the margin based on the norm of the Jacobian with respect to the input layer. They show that maximizing their approximations to the margin leads to improved generalization. However, their analysis was restricted to margin at the input layer. Poggio et al. (2017) and Liao et al. (2018) propose a normalized cross-entropy measure that correlates well with test loss. Their proposed normalized loss trades off confidence of predictions with stability, which leads to better correlation with test accuracy, leading to a significant lowering of output margin.

## 3 PREDICTION OF GENERALIZATION GAP

In this section, we introduce our margin-based measure. We first explain the construction scheme for obtaining the margin distribution. We then squeeze the distributional information of the margin to a small number of statistics. Finally, we regress these statistics to the value of the generalization gap. We assess prediction quality by applying the learned regression coefficients to predict the generalization gap of unseen models. We will start with providing a motivation for using the margins at the hidden layers which is supported by our empirical findings. SVM owes a large part of its success to the kernel that allows for inner product in a higher and richer feature space. At its crux, the primal kernel SVM problem is separated into the feature extractor and the classifier on the extracted features. We can separate any feed forward network at any given hidden layer and treat the hidden representation as a feature map. From this view, the layers that precede this hidden layer can be treated as a learned feature extractor and then the layers that come after are naturally the classifier. If the margins at the input layers or the output layers play important roles in generalization of the classifier, it is a natural conjecture that the margins at these hidden representations are also important in generalization. In fact, if we ignore the optimization procedure and focus on a converged network, generalization theories developed on the input such as Lv et al. (2019) can be easily extended to the hidden layers or the extracted features.

### 3.1 MARGIN APPROXIMATION

First, we establish some notation. Consider a classification problem with $n$ classes. We assume a classifier $f$ consists of non-linear functions $f_i : \mathcal{X} \to \mathbb{R}$, for $i = 1, \ldots, n$ that generate a prediction score for classifying the input vector $\boldsymbol{x} \in \mathcal{X}$ to class $i$. The predicted label is decided by the class with maximal score, i.e. $i^* = \arg\max_i f_i(\boldsymbol{x})$. Define the *decision boundary* for each class pair $(i, j)$

as:

$$\mathcal{D}_{(i,j)} \triangleq \{\boldsymbol{x} \mid f_i(\boldsymbol{x}) = f_j(\boldsymbol{x})\} \tag{1}$$

Under this definition, the $l_p$ distance of a point $\boldsymbol{x}$ to the decision boundary $\mathcal{D}_{(i,j)}$ can be expressed as the smallest displacement of the point that results in a score tie:

$$d_{f,\boldsymbol{x},(i,j)} \triangleq \min_{\boldsymbol{\delta}} \|\boldsymbol{\delta}\|_p \quad \text{s.t.} \quad f_i(\boldsymbol{x} + \boldsymbol{\delta}) = f_j(\boldsymbol{x} + \boldsymbol{\delta}) \tag{2}$$

Unlike an SVM, computing the "exact" distance of a point to the decision boundary (Eq. 2) for a deep network is intractable[3]. In this work, we adopt the approximation scheme from Elsayed et al. (2018) to capture the distance of a point to the decision boundary. This a first-order Taylor approximation to the true distance Eq. 2. Formally, given an input $\boldsymbol{x}$ to a network, denote its representation at the $l^{th}$ layer (the layer activation vector) by $\boldsymbol{x}^l$. For the input layer, let $l = 0$ and thus $\boldsymbol{x}^0 = \boldsymbol{x}$. Then for $p = 2$, the distance of the representation vector $\boldsymbol{x}^l$ to the decision boundary for class pair $(i,j)$ is given by the following approximation:

$$d_{f,(i,j)}(\boldsymbol{x}^l) = \frac{f_i(\boldsymbol{x}^l) - f_j(\boldsymbol{x}^l)}{\|\nabla_{\boldsymbol{x}^l} f_i(\boldsymbol{x}^l) - \nabla_{\boldsymbol{x}^l} f_j(\boldsymbol{x}^l)\|_2} \tag{3}$$

Here $f_i(\boldsymbol{x}^l)$ represents the output (logit) of the network logit $i$ given $\boldsymbol{x}^l$. Note that this distance can be positive or negative, denoting whether the training sample is on the "correct" or "wrong" side of the decision boundary respectively. This distance is well defined for all $(i,j)$ pairs, but in this work we assume that $i$ always refers to the ground truth label and $j$ refers to the second highest or highest class (if the point is misclassified). The training data $\boldsymbol{x}$ induces a distribution of distances at each layer $l$ which, following earlier naming convention (Garg et al., 2002; Langford & Shawe-Taylor, 2002), we refer to as *margin distribution* (at layer $l$). For margin distribution, we only consider distances with positive sign (we ignore all misclassified training points). Such design choice facilitates our empirical analysis when we transform our features (e.g. log transform); further, it has also been suggested that it may be possible to obtain a better generalization bound by only considering the correct examples when the classifier classifies a significant proportion of the training examples correctly, which is usually the case for neural networks (Bartlett, 1998). For completeness, the results with negative margins are included in appendix Sec. 7.

A problem with plain distances and their associated distribution is that they can be trivially boosted without any significant change in the way classifier separates the classes. For example, consider multiplying weights at a layer by a constant and dividing weights in the following layer by the same constant. In a ReLU network, due to positive homogeneity property (Liao et al., 2018), this operation does not affect how the network classifies a point, but it changes the distances to the decision boundary[4]. To offset the scaling effect, we normalize the margin distribution. Consider margin distribution at some layer $l$, and let $\boldsymbol{x}_k^l$ be the representation vector for training sample $k$. We compute the variance of each coordinate of $\{\boldsymbol{x}_k^l\}$ separately, and then sum these individual variances. This quantity is called *total variation* of $\boldsymbol{x}^l$. The square root of this quantity relates to the scale of the distribution. That is, if $\boldsymbol{x}^l$ is scaled by a factor, so is the square root of the total variation. Thus, by dividing distances by the square root of total variation, we can construct a margin distribution invariant to scaling. More concretely, the total variation is computed as:

$$\nu(\boldsymbol{x}^l) = \text{tr}\left(\frac{1}{n}\sum_{k=1}^{n}(\boldsymbol{x}_k^l - \bar{\boldsymbol{x}}^l)(\boldsymbol{x}_k^l - \bar{\boldsymbol{x}}^l)^T\right) \quad , \quad \bar{\boldsymbol{x}}^l = \frac{1}{n}\sum_{k=1}^{n}\boldsymbol{x}_k^l \, , \tag{4}$$

i.e. the trace of the empirical covariance matrix of activations. Using the total variation, the normalized margin is specified by:

$$\hat{d}_{f,(i,j)}(\boldsymbol{x}_k^l) = \frac{d_{f,(i,j)}(\boldsymbol{x}_k^l)}{\sqrt{\nu(\boldsymbol{x}^l)}} \tag{5}$$

While the quantity is relatively primitive and easy to compute, Fig. 1 (top) shows that the normalized-margin distributions based on Eq. 5 have the desirable effect of becoming heavier tailed and shifting

---

[3]This is because computing the distance of a point to a nonlinear surface is intractable. This is different from SVM where the surface is linear and distance of a point to a hyperplane admits a closed form expression.

[4]For example, suppose the constant $c$ is greater that one. Then, multiplying the weights of a layer by $c$ magnifies distances computed at the layer by a factor of $c$.

to the right (increasing margin) as generalization gap decreases. We find that this effect holds across a range of networks trained with different hyper-parameters.

## 3.2 Summarizing the Margin Distribution

Instead of working directly with the (normalized) margin distribution, it is easier to analyze a compact *signature* of that. The moments of a distribution are a natural criterion for this purpose. Perhaps the most standard way of doing this is computing the empirical moments from the samples and then take the $n^{th}$ root of the $n^{th}$ moment. In our experiments, we used the first five moments. However, it is a well-known phenomenon that the estimation of higher order moments based on samples can be unreliable. Therefore, we also consider an alternate way to construct the distribution's signature. Given a set of distances $\mathcal{D} = \{\hat{d}_m\}_{m=1}^n$, which constitute the margin distribution. We use the median $Q_2$, first quartile $Q_1$ and third quartile $Q_3$ of the normalized margin distribution, along with the two *fences* that indicate variability outside the upper and lower quartiles. There are many variations for fences, but in this work, with $IQR = Q_3 - Q_1$, we define the upper fence to be $\max(\{\hat{d}_m : \hat{d}_m \in \mathcal{D} \wedge \hat{d}_m \leq Q_3 + 1.5IQR\})$ and the lower fence to be $\min(\{\hat{d}_m : \hat{d}_m \in \mathcal{D} \wedge \hat{d}_m \geq Q_1 - 1.5IQR\})$ (McGill et al., 1978). These 5 statistics form the *quartile* description that summarizes the normalized margin distribution at a specific layer, as shown in the box plots of Fig. 1. We will later see that both signature representations are able to predict the generalization gap, with the second signature working slightly better.

A number of prior works such as Bartlett et al. (2017), Neyshabur et al. (2017b), Liu et al. (2016), Sun et al. (2015), Sokolic et al. (2016), and Liang et al. (2017) have focused on analyzing or maximizing the margin at either the input or the output layer of a deep network. Since a deep network has many hidden layers with evolving representations, it is not immediately clear which of the layer margins is of importance for improving generalization. Our experiments reveal that margin distribution from all of the layers of the network contribute to prediction of generalization gap. This is also clear from Fig. 1 (top): comparing the input layer (layer 0) margin distributions between the left and right plots, the input layer distribution shifts slightly left, but the other layer distributions shift the other way. For example, if we use quartile signature, we have $5L$ components in this vector, where $L$ is the total number of layers in the network. We incorporate dependence on all layers simply by concatenating margin signatures of all layers into a single combined vector $\boldsymbol{\theta}$ that we refer to as *total signature*. Empirically, we found constructing the total signature based on four evenly-spaced layers (*input, and 3 hidden layers*) sufficiently captures the variation in the distributions and generalization gap, and also makes the signature *agnostic* to the depth of the network.

## 3.3 Evaluation Metrics

Our goal is to predict the generalization gap, i.e. the *difference between training and test accuracy* at the end of training, based on total signature $\boldsymbol{\theta}$ of a trained model. We use the simplest prediction model, i.e. a linear form $\hat{g} = \boldsymbol{a}^T \phi(\boldsymbol{\theta}) + b$, where $\boldsymbol{a} \in \mathbb{R}^{\dim(\boldsymbol{\theta})}$ and $b \in \mathbb{R}$ are parameters of the predictor, and $\phi : \mathbb{R} \rightarrow \mathbb{R}$ is a function applied element-wise to $\boldsymbol{\theta}$. Specifically, we will explore two choices of $\phi$: the identity $\phi(x) = x$ and entry-wise log transform $\phi(x) = \log(x)$, which correspond to additive and multiplicative combination of margin statistics respectively. We do *not* claim this model is the true relation, but rather it is a simple model for prediction; and our results suggest that it is a surprisingly good approximation.

In order to estimate predictor parameters $\boldsymbol{a}, b$, we generate a pool of $n$ pretrained models (covering different datasets, architectures, regularization schemes, etc. as explained in Sec. 4) each of which gives one instance of the pair $\boldsymbol{\theta}, g$ ($g$ being the generalization gap for that model). We then find $\boldsymbol{a}, b$ by minimizing mean squared error: $(\boldsymbol{a}^*, b^*) = \arg\min_{\boldsymbol{a},b} \sum_i (\boldsymbol{a}^T \phi(\boldsymbol{\theta}_i) + b - g_i)^2$, where $i$ indexes the $i^{th}$ model in the pool. The next step is to assess the prediction quality. We consider two metrics for this. The first metric examines quality of predictions on unseen models. For that, we consider a held-out pool of $m$ models, different from those used to estimate $(\boldsymbol{a}, b)$, and compute the value of $\hat{g}$ on them via $\hat{g} = \boldsymbol{a}^T \phi(\boldsymbol{\theta}) + b$. In order to quantify the discrepancy between predicted gap $\hat{g}$ and ground truth gap $g$ we use the notion of *coefficient of determination* ($R^2$) (Glantz et al., 1990):

$$R^2 = 1 - \frac{\sum_{j=1}^n (\hat{g}_j - g_j)^2}{\sum_{j=1}^n (g_j - \frac{1}{n}\sum_{j=1}^n g_j)^2} \tag{6}$$

$R^2$ measures what fraction of data variance can be explained by the linear model[5] (it ranges from 0 to 1 on training points but can be outside that range on unseen points). To be precise, we use k-fold validation to study how the predictor can perform on held out pool of trained deep networks. We use 90/10 split, fit the linear model with the training pool, and measure $R^2$ on the held out pool. The performance is averaged over the 10 splits. Since $R^2$ is now not measured on the training pool, it does not suffer from high data dimension and can be negative. In all of our experiments, we use $k = 10$. We provide a subset of residual plots and corresponding univariate F-Test for the experiments in the appendix (Sec. 8). The F-score also indicates how important each individual variable is. The second metric examines how well the model fits based on the provided training pool; it does not require a test pool. To characterize this, we use *adjusted* $\bar{R}^2$ (Glantz et al., 1990) defined as:

$$\bar{R}^2 = 1 - (1 - R^2)\frac{n - 1}{n - \dim(\boldsymbol{\theta}) - 1} \, . \tag{7}$$

The $\bar{R}^2$ can be negative when the data is non-linear. Note that $\bar{R}^2$ is always smaller than $R^2$. Intuitively, $\bar{R}^2$ penalizes the model if the number of features is high relative to the available data points. The closer $\bar{R}^2$ is to 1, the better the model fits. Using $\bar{R}^2$ is a simple yet effective method to test the fitness of linear model and is independent of the scale of the target, making it a more illustrative metric than residuals.

## 4 EXPERIMENTS

We tested our measure of generalization gap $\hat{g}$, along with baseline measures, on a number of deep networks and architectures: nine-layer convolutional networks on CIFAR-10 (10 with input layer), and 32-layer residual networks on both CIFAR-10 and CIFAR-100 datasets. The trained models and relevant Tensorflow Abadi et al. (2016) code to compute margin distributions are released at https://github.com/google-research/google-research/tree/master/demogen

### 4.1 CONVOLUTIONAL NEURAL NETWORKS ON CIFAR-10

Using the CIFAR-10 dataset, we train 216 nine-layer convolutional networks with different settings of hyperparameters and training techniques. We apply weight decay and dropout with different strengths; we use networks with and without batch norm and data augmentation; we change the number of hidden units in the hidden layers. Finally, we also include training with and without corrupted labels, as introduced in Zhang et al. (2016); we use a fixed amount of $20\%$ corruption of the true labels. The *accuracy* on the test set ranges from $60\%$ to $90.5\%$ and the *generalization gap* ranges from $1\%$ to $35\%$. In standard settings, creating neural network models with small generalization gap is difficult; in order to create sufficiently diverse generalization behaviors, we limit some models' capacities by large weight regularization which decreases generalization gap by lowering the training accuracy. All networks are trained by SGD with momentum. Further details are provided in the supplementary material (Sec. 6).

For each trained network, we compute the depth-agnostic signature of the normalized margin distribution (see Sec. 3). This results in a 20-dimensional signature vector. We estimate the parameters of the linear predictor $(\boldsymbol{a}, b)$ with the log transform $\phi(x) = \log(x)$ and using the 20-dimensional signature vector $\boldsymbol{\theta}$. Fig. 2 (left) shows the resulting scatter plot of the predicted generalization gap $\hat{g}$ and the true generalization gap $g$. As it can be seen, it is very close to being linear across the range of generalization gaps, and this is also supported by the $\bar{R}^2$ of the model, which is 0.96 (max is 1).

As a first baseline method, we compare against the work of Bartlett et al. (2017) which provides one of the best generalization bounds currently known for deep networks. This work also constructs a margin distribution for the network, but in a different way. To make a fair comparison, we extract the same signature $\boldsymbol{\theta}$ from their margin distribution. Since their margin distribution can only be defined for the output layer, their $\boldsymbol{\theta}$ is 5-dimensional for any network. The resulting fit is shown in Fig. 2(right). It is clearly a poorer fit than that of our signature, with a significantly lower $\bar{R}^2$ of 0.72.

For a fairer comparison, we also reduced our signature $\boldsymbol{\theta}$ from 20 dimensions to the best performing 4 dimensions (even one dimension less than what we used for Bartlett's) by dropping 16 components in our $\boldsymbol{\theta}$. This is shown in Fig. 2 (middle) and has a $\bar{R}^2$ of 0.89, which is poorer than our complete

---

[5] A simple manipulation shows that the prediction residual $\sum_{j=1}^{m}(\hat{g}_j - g_j)^2 \propto 1 - R^2$, so $R^2$ can be interpreted as a *scale invariant* alternative to the residual.

| Experiment Settings | CNN+CIFAR10 | | | ResNet+CIFAR10 | | | ResNet+CIFAR100 | | |
|---|---|---|---|---|---|---|---|---|---|
| | A $R^2$ | kf $R^2$ | mse | A $R^2$ | kf $R^2$ | mse | A $R^2$ | kf $R^2$ | mse |
| `qrt+log` | **0.94** | **0.90** | **1.5** | **0.87** | **0.81** | **0.40** | **0.97** | **0.96** | **1.6** |
| `qrt+log+unnorm` | 0.89 | 0.86 | 2.0 | 0.82 | 0.74 | 0.48 | 0.95 | 0.94 | 1.9 |
| `qrt+linear` | 0.88 | 0.84 | 2.2 | 0.77 | 0.66 | 0.54 | 0.91 | 0.87 | 2.6 |
| `sf+log` | 0.73 | 0.69 | 3.5 | 0.53 | 0.41 | 0.80 | 0.80 | 0.78 | 4.0 |
| `sl+log` | 0.86 | 0.84 | 2.2 | 0.44 | 0.33 | 0.87 | 0.95 | 0.94 | 1.8 |
| `moment+log` | 0.93 | 0.87 | 1.6 | 0.83 | 0.74 | 0.45 | 0.94 | 0.92 | 2.0 |
| `best4+log` | 0.89 | 0.88 | 2.1 | 0.54 | 0.43 | 0.80 | 0.93 | 0.92 | 2.4 |
| `spectral+log` | 0.73 | 0.70 | 3.5 | - | - | - | - | - | - |
| `Jacobian+log` | 0.42 | N | 5.0 | 0.20 | N | 1.0 | 0.47 | N | 6.0 |
| `LM+linear` | 0.35 | N | 5.2 | 0.68 | N | 0.66 | 0.74 | N | 4.0 |

Table 1: Ablation experiments considering different scenarios (see text for details). The last 3 rows are baselines from: (Bartlett et al., 2017; Sokolic et al., 2016; Elsayed et al., 2018). **A** indicates adjusted; **kf** indicates k-fold; **mse** indicates mean squared error in $10^{-3}$; N indicates negative.

$\theta$ but still significantly higher than that of Bartlett et al. (2017). In addition, we considered two other baseline comparisons: Sokolic et al. (2016), where margin at input is defined as a function of the Jacobian of output (logits) with respect to input; and Elsayed et al. (2018) where the linearized approximation to margin is derived (for the same layers where we use our normalized margin approximation).

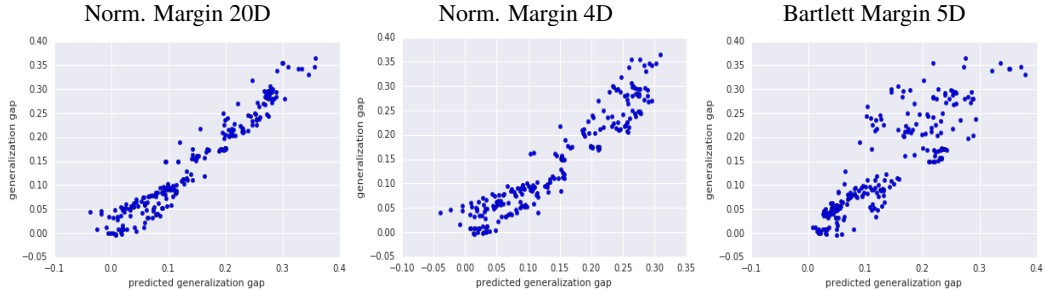

Figure 2: (Best seen as PDF) Regression models to predict generalization gap. Left: regression model fit in log space for the full 20-dimensional feature space ($\bar{R}^2 = 0.94$); Middle: fit for a subset of only 4 features, 2 each from 2 of the hidden layers ($\bar{R}^2 = 0.89$); Right: fit for features extracted from the normalized margin distribution as used in Bartlett et al. (2017) ($\bar{R}^2 = 0.72$).

To quantify the effect of the normalization, different layers, feature transformation etc., we conduct a number of ablation experiments with the following configuration: 1. `linear`/`log`: Use signature transform of $\phi(x) = x$ or $\phi(x) = \log(x)$; 2. `sl`: Use signature from the single best layer ($\boldsymbol{\theta} \in \mathbb{R}^5$); 3. `sf`: Use only the single best statistic from the total signature for all the layers ($\boldsymbol{\theta} \in \mathbb{R}^4$, individual layer result can be found in Sec. 7); 4. `moment`: Use the first 5 moments of the normalized margin distribution as signature instead of quartile statistics $\boldsymbol{\theta} \in \mathbb{R}^{20}$ (Sec. 3); 5. `spectral`: Use signature of spectrally normalized margins from Bartlett et al. (2017) ($\boldsymbol{\theta} \in \mathbb{R}^5$); 6. `qrt`: Use all the quartile statistics as total signature $\boldsymbol{\theta} \in \mathbb{R}^{20}$ (Sec. 3); 7. `best4`: Use the 4 best statistics from the total signature ($\boldsymbol{\theta} \in \mathbb{R}^4$); 8. `Jacobian`: Use the Jacobian-based margin defined in Eq (39) of Sokolic et al. (2016) ($\boldsymbol{\theta} \in \mathbb{R}^5$); 9. `LM`: Use the large margin loss from Elsayed et al. (2018) at the same four layers where the statistics are measured ($\boldsymbol{\theta} \in \mathbb{R}^4$); 10. `unnorm` indicates no normalization. In Table 1, we list the $\bar{R}^2$ from fitting models based on each of these scenarios. We see that, both quartile and moment signatures perform similarly, lending support to our thesis that the margin distribution, rather than the smallest or largest margin, is of importance in the context of generalization.

## 4.2 RESIDUAL NETWORKS ON CIFAR-10

On the CIFAR-10 dataset, we train 216 convolutional networks with residual connections; these networks are 32 layers deep with standard ResNet 32 topology (He et al., 2016). Since it is difficult to train ResNet without activation normalization, we created generalization gap variation with batch

normalization (Ioffe & Szegedy, 2015) and group normalization (Wu & He, 2018). We further use different initial learning rates. The range of accuracy on the test set ranges from $83\%$ to $93.5\%$ and generalization gap from $6\%$ to $13.5\%$. The residual networks were much deeper, and so we only chose 4 layers for feature-length compatibility with the shallower convoluational networks. This design choice also facilitates ease of analysis and circumvents the dependency on depth of the models. Table 1 shows the $\bar{R}^2$. Note in the presence of residual connections that use convolution instead of identity and identity blocks that span more than one convolutional layers, it is not immediately clear how to properly apply the bounds of Bartlett et al. (2017) (third from last row) without morphing the topology of the architecture and careful design of reference matrices. As such, we omit them for ResNet. Fig. 3 (left) shows the fit for the resnet models, with $\bar{R}^2 = 0.87$. Fig. 3 (middle) and Fig. 3 (right) compare the log normalized density plots of a CIFAR-10 resnet and CIFAR-10 CNN. The plots show that the Resnet achieves a better margin distribution, correlated with greater test accuracy, even though it was trained without data augmentation.

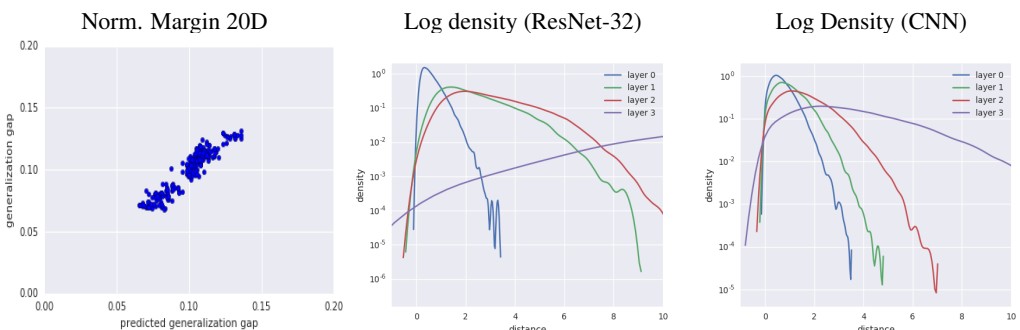

Figure 3: (Best seen as PDF) Left: Regression model fit in log space for the full 20-dimensional feature space for 216 residual networks ($\bar{R}^2 = 0.87$) on CIFAR-10; Middle: Log density plot of normalized margins of a particular residual network that achieves $91.7\%$ test accuracy without data augmentation; Right: Log density plot of normalized margins of a CNN that achieves $87.2\%$ with data augmentation. We see that the resnet achieves larger margins, especially at the hidden layers, and this is reflected in the higher test accuracy.

### 4.3 RESNET ON CIFAR-100

On the CIFAR-100 dataset, we trained 324 ResNet-32 with the same variation in hyperparameter settings as for the networks for CIFAR-10 with one additional initial learning rate. The range of accuracy on the test set ranges from $12\%$ to $73\%$ and the generalization gap ranges from $1\%$ to $75\%$. Table 1 shows $\bar{R}^2$ for a number of ablation experiments and the full feature set. Fig. 4 (left) shows the fit of predicted and true generalization gaps over the networks ($\bar{R}^2 = 0.97$). Fig. 4 (middle) and Fig. 4 (right) compare a CIFAR-100 residual network and a CIFAR-10 residual network with the same architecture and hyperparameters. Under these settings, the CIFAR-100 network achieves $44\%$ test accuracy, whereas CIFAR-10 achieves $61\%$. The resulting normalized margin density plots clearly reflect the better generalization achieved by CIFAR-10: the densities at all layers are wider and shifted to the right. Thus, the normalized margin distributions reflect the relative "difficulty" of a particular dataset for a given architecture.

## 5 DISCUSSION

We have presented a predictor for generalization gap based on margin distribution in deep networks and conducted extensive experiments to assess it. Our results show that our scheme achieves a high adjusted coefficient of determination (a linear regression predicts generalization gap accurately). Specifically, the predictor uses normalized margin distribution across multiple layers of the network. The best predictor uses quartiles of the distribution combined in multiplicative way (additive in $\log$ transform). Compared to the strong baseline of spectral complexity normalized output margin (Bartlett et al., 2017), our scheme exhibits much higher predictive power and can be applied to any feedforward network (including ResNets, unlike generalization bounds such as (Bartlett et al., 2017; Neyshabur et al., 2017b; Arora et al., 2018)). We also find that using hidden layers is crucial for the predictive power. Our findings could be a stepping stone for studying new generalization theories and

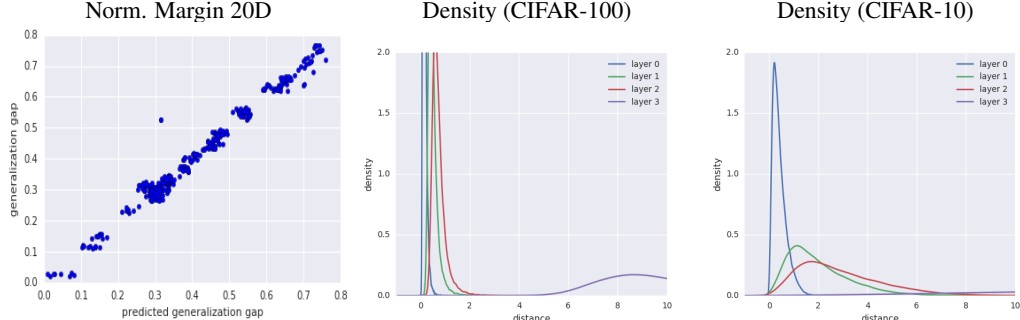

Figure 4: (Best seen as PDF) Left: Regression model fit in log space for the full 20-dimensional feature space for 300 residual networks ($\bar{R}^2 = 0.97$) on CIFAR-100; Middle: density plot of normalized margins of a particular residual network trained on CIFAR-100 that achieves $44\%$ test accuracy; Right: Density plot of normalized margins of a residual network trained on CIFAR-10 that achieves $61\%$.

new loss functions with better generalization properties. We include the results on cross architecture and cross data comparison as well as some final thoughts in Appendix Sec. 9.

ACKNOWLEDGMENTS

We are thankful to Gamaleldin Elsayed (Google), Tomer Koren (Google), Sergey Ioffe (Google), Vighnesh Birodkar (Google), Shraman Ray Chaudhuri (Google), Kevin Regan (Google), Behnam Neyshabur (NYU), and Dylan Foster (Cornell), for discussions and helpful feedbacks.

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

## 6 APPENDIX: EXPERIMENTAL DETAILS

### 6.1 CNN + CIFAR-10

We use an architecture very similar to Network in Network (Lin et al. (2013)), but we remove all dropout and max pool from the network.

| Layer Index | Layer Type | Output Shape |
|:---:|:---:|:---:|
| 0 | Input | $32 \times 32 \times 3$ |
| 1 | $3 \times 3$ convolution + stride 2 | $16 \times 16 \times 192$ |
| 2 | $1 \times 1$ convolution + stride 1 | $16 \times 16 \times 192$ |
| 3 | $1 \times 1$ convolution + stride 1 | $16 \times 16 \times 192$ |
| 4 | $3 \times 3$ convolution + stride 2 | $8 \times 8 \times 192$ |
| 5 | $1 \times 1$ convolution + stride 1 | $8 \times 8 \times 192$ |
| 6 | $1 \times 1$ convolution + stride 1 | $8 \times 8 \times 192$ |
| 7 | $3 \times 3$ convolution + stride 2 | $4 \times 4 \times 192$ |
| 8 | $1 \times 1$ convolution + stride 1 | $4 \times 4 \times 192$ |
| 9 | $1 \times 1$ convolution + stride 1 | $4 \times 4 \times 192$ |
| 10 | $4 \times 4$ convolution + stride 1 | $1 \times 1 \times 10$ |

Table 2: Architecture of base CNN model.

To create generalization gap in this model, we make the following modification to the base architecture:

1. Use channel size of 192, 288, and 384 to create different width

2. Train with and without batch norm at all convolutional layers

3. Apply dropout at layer 3 and 6 with $p = 0.0, 0.2, 0.5$

4. Apply $l_2$ regularization with $\lambda = 0.0, 0.001, 0.005$

5. Train with and without data augmentation with random cropping, flipping and shifting

6. Train each configuration twice

In total this gives us $3 \times 2 \times 3 \times 3 \times 2 \times 2 = 216$ different network architectures. The models are trained with SGD with momentum ($\alpha = 0.9$) at minibatch size of 128 and intial learning rate of 0.01. All networks are trained for 380 epoch with $10\times$ learning rate decay at interval of 100 epoch.

### 6.2 RESNET 32 + CIFAR-10

For this experiments, we use the standard ResNet 32 architectures. We consider down sampling to the marker of a stage, so there are in total 3 stages in the ResNet 32 architecture. To create generalization gap in this model, we make the following modifications to the architecture:

1. Use network width that are $1\times, 2\times, 4\times$ wider in number of channels.

2. Train with batch norm or group norm (Wu & He, 2018)

3. Train with initial learning rate of $0.01, 0.001$

4. Apply $l_2$ regularization with $\lambda = 0.0, 0.02, 0.002$

5. Trian with and without data augmentation with random cropping, flipping and shifting

6. Train each configuration 3 times

In total this gives us $3 \times 2 \times 2 \times 3 \times 2 \times 3 = 216$ different network architectures. The models are trained with SGD with momentum ($\alpha = 0.9$) at minibatch size of 128. All networks are trained for 380 epoch with $10\times$ learning rate decay at interval of 100 epoch.

### 6.3 RESNET 32 + CIFAR-100

For this experiments, we use the standard ResNet 32 architectures. We consider down sampling to the marker of a stage, so there are in total 3 stages in the ResNet 32 architecture. To create generalization gap in this model, we make the following modifications to the architecture:

1. Use network width that are $1\times, 2\times, 4\times$ wider in number of channels.
2. Train with batch norm or group norm (Wu & He, 2018)
3. Train with initial learning rate of $0.1, 0.01, 0.001$
4. Apply $l_2$ regularization with $\lambda = 0.0, 0.02, 0.002$
5. Trian with and without data augmentation with random cropping, flipping and shifting
6. Train each configuration 3 times

In total this gives us $3 \times 2 \times 3 \times 3 \times 2 \times 3 = 324$ different network architectures. The models are trained with SGD with momentum ($\alpha = 0.9$) at minibatch size of 128. All networks are trained for 380 epoch with $10\times$ learning rate decay at interval of 100 epoch.

## 7 APPENDIX: MORE REGRESSION RESULTS

### 7.1 ANALYSIS WITH NEGATIVE MARGINS

The last two rows contain the results of including the negative margins and regress against both the gap (generalization gap) and against acc (test accuracy). We see that including when negative margin is included, it is in general easier to predict the accuracy of the models rather than the gap itself. For convenience, we have reproduced Table 1.

| Experiment Settings | CNN+CIFAR10 | | | ResNet+CIFAR10 | | | ResNet+CIFAR100 | | |
|---|---|---|---|---|---|---|---|---|---|
| | **A $R^2$** | **kf $R^2$** | **mse** | **A $R^2$** | **kf $R^2$** | **mse** | **A $R^2$** | **kf $R^2$** | **mse** |
| qrt+log | **0.94** | **0.90** | **1.5** | **0.87** | **0.81** | **0.40** | **0.97** | **0.96** | **1.6** |
| qrt+log+unnorm | 0.89 | 0.86 | 2.0 | 0.82 | 0.74 | 0.48 | 0.95 | 0.94 | 1.9 |
| qrt+linear | 0.88 | 0.84 | 2.2 | 0.77 | 0.66 | 0.54 | 0.91 | 0.87 | 2.6 |
| sf+log | 0.73 | 0.69 | 3.5 | 0.53 | 0.41 | 0.80 | 0.80 | 0.78 | 4.0 |
| sl+log | 0.86 | 0.84 | 2.2 | 0.44 | 0.33 | 0.87 | 0.95 | 0.94 | 1.8 |
| moment+log | 0.93 | 0.87 | 1.6 | 0.83 | 0.74 | 0.45 | 0.94 | 0.92 | 2.0 |
| best4+log | 0.89 | 0.88 | 2.1 | 0.54 | 0.43 | 0.80 | 0.93 | 0.92 | 2.4 |
| spectral+log | 0.73 | 0.70 | 3.5 | - | - | - | - | - | - |
| Jacobian+log | 0.42 | N | 5.0 | 0.20 | N | 1.0 | 0.47 | N | 6.0 |
| LM+linear | 0.35 | N | 5.2 | 0.68 | N | 0.66 | 0.74 | N | 4.0 |
| qrt+linear+gap | 0.90 | 0.86 | 2.0 | 0.71 | 0.61 | 0.6 | 0.93 | 0.89 | 2.0 |
| qrt+linear+acc | 0.91 | 0.88 | 1.6 | 0.98 | 0.96 | 0.2 | 0.92 | 0.89 | 2.0 |

Table 3: Table 1 with two additional row. The second to last row shows prediction of generalization gap when negative margins are used; the last row is for prediction of test *accuracy* rather than gap. Note that negative margins can only be used with linear features.

### 7.2 ANALYSIS FOR INDIVIDUAL LAYER'S MARGIN DISTRIBUTIONS

This is a comparison of different individual layer's predictive power by using only the margin distribution at that layer. This results illustrates the importance of margins in the hidden layers.

| Experiment Settings | CNN+CIFAR10 | | | ResNet+CIFAR10 | | | ResNet+CIFAR100 | | |
|---|---|---|---|---|---|---|---|---|---|
| | **A $R^2$** | **kf $R^2$** | **mse** | **A $R^2$** | **kf $R^2$** | **mse** | **A $R^2$** | **kf $R^2$** | **mse** |
| input | 0.77 | 0.73 | 3.1 | 0.16 | 0.02 | 1.0 | 0.82 | 0.81 | 3.9 |
| h1 | 0.77 | 0.74 | 3.1 | 0.36 | 0.26 | 0.94 | 0.95 | 0.94 | 1.8 |
| h2 | 0.80 | 0.77 | 3.0 | 0.41 | 0.31 | 0.90 | 0.77 | 0.73 | 4.4 |
| h3 | 0.86 | 0.84 | 2.2 | 0.44 | 0.33 | 0.87 | 0.54 | 0.53 | 6.3 |
| all layer | **0.94** | **0.90** | **1.5** | **0.87** | **0.81** | **0.40** | **0.97** | **0.96** | **1.6** |

Table 4: Single layer comparison, all with qrt+log

# 8 APPENDIX: FURTHER ANALYSIS OF REGRESSION

## 8.1 CNN + CIFAR-10 + ALL QUARTILE SIGNATURE

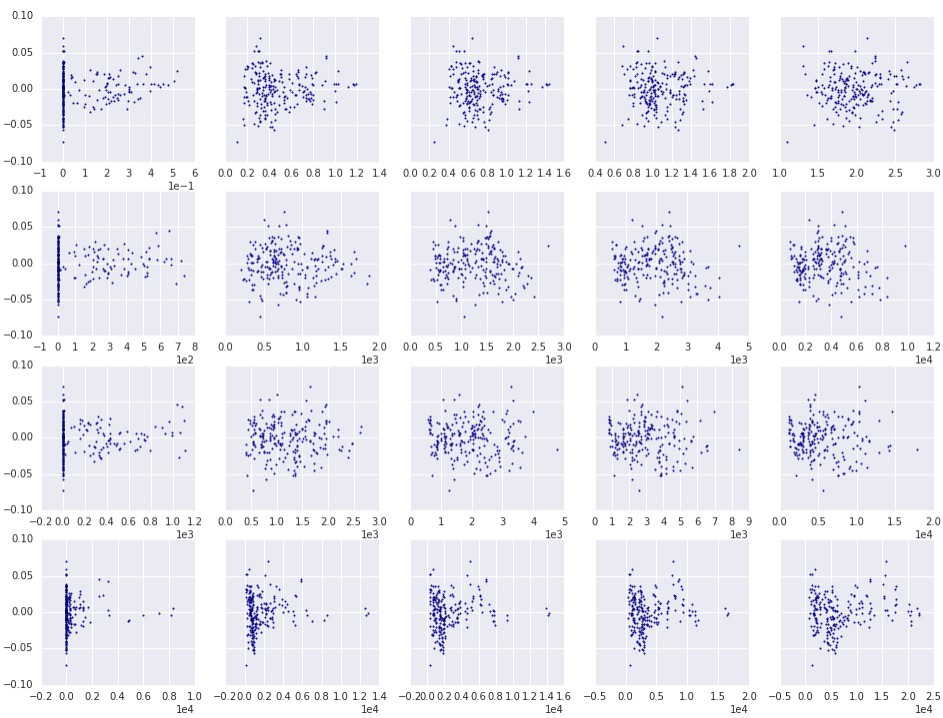

Figure 5: Residual plots for all explanatory variables, row: h0, h1, h2, h3, column: lower fence, $Q_1$, $Q_2$, $Q_3$, upper fence. lower fence is clipped because distance cannot be smaller than 0. The residual is fairly evenly distributed around 0.

|      | lower fence | $Q_1$     | $Q_2$     | $Q_3$     | upper fence |
|------|-------------|-----------|-----------|-----------|-------------|
| h0   | 306.40      | 114.41    | 39.56     | 12.54     | 5.07        |
| h1   | 286.53      | 9.42      | 5.16      | 17.29     | 38.57       |
| h2   | 259.68      | 6.95      | 77.03     | 110.40    | 152.20      |
| h3   | 188.59      | 10.29     | 49.76     | 83.40     | 143.23      |
|      | lower fence | $Q_1$     | $Q_2$     | $Q_3$     | upper fence |
| h0   | 3.59e-43    | 1.13e-21  | 1.76e-9   | 4.87e-4   | 2.52e-2     |
| h1   | 2.34e-41    | 2.41e-3   | 2.40e-2   | 4.64e-5   | 2.70e-09    |
| h2   | 8.76e-39    | 8.95e-3   | 5.38e-16  | 4.30e-21  | 9.12e-27    |
| h3   | 3.40e-31    | 1.54e-3   | 2.37e-11  | 5.17e-17  | 1.31e-25    |

Table 5: F score (top) and p-values (bottom) for all 20 variables. Using $p = 0.05$, the null hypotheses are rejected for every variable.

## 8.2 ResNet 32 + CIFAR-10 + All Quartile Signature

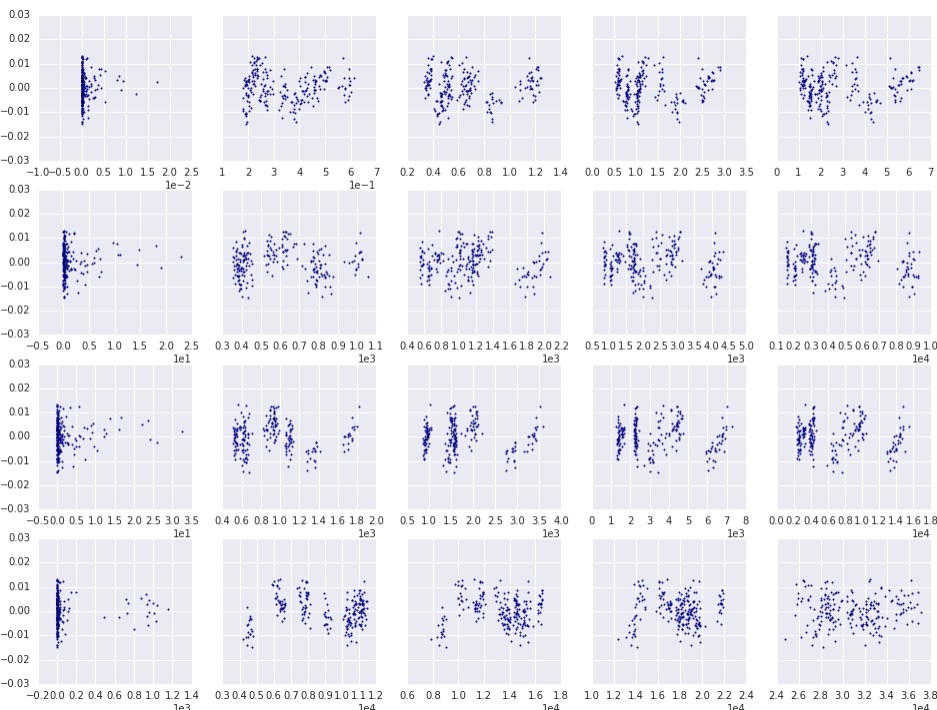

Figure 6: Residual plots for all explanatory variables, row: h0, h1, h2, h3, column: lower fence, $Q_1$, $Q_2$, $Q_3$, upper fence. lower fence is clipped because distance cannot be smaller than 0. The residual is less evenly distributed as are in other two settings; this fact is well reflected in the cluster along the x axis and in the $\bar{R}^2$; we speculate that this is due to not having diverse enough generalization gap in the models trained to cover the entire space of the "model" unlike in the other two settings.

|     | lower fence | $Q_1$ | $Q_2$ | $Q_3$ | upper fence |
|-----|-------------|-------|-------|-------|-------------|
| h0  | 45.67       | 16.67 | 6.97  | 1.71  | 0.68        |
| h1  | 58.84       | 88.14 | 44.15 | 15.59 | 9.36        |
| h2  | 60.20       | 78.57 | 35.76 | 12.89 | 7.52        |
| h3  | 59.75       | 0.27  | 1.192 | 7.37  | 44.22       |

|     | lower fence | $Q_1$    | $Q_2$    | $Q_3$    | upper fence |
|-----|-------------|----------|----------|----------|-------------|
| h0  | 1.30e-10    | 6.25e-5  | 8.88e-3  | 0.192    | 0.40        |
| h1  | 5.94e-13    | 9.33e-18 | 2.47e-10 | 1.06e-4  | 2.49e-3     |
| h2  | 3.45e-13    | 3.04e-16 | 9.21e-9  | 4.07e-4  | 6.59e-3     |
| h3  | 4.14e-13    | 0.60     | 0.27     | 7.14e-3  | 2.4e-10     |

Table 6: F score (top) and p-values (bottom) for all 20 variables. Using $p = 0.05$, we see that the null hypotheses are not rejected for 4 of the variables. We believe having a more diverse generalization behavior in the study will solve this problem.

## 8.3 ResNet 32 + CIFAR-100 + All Quartile Signature

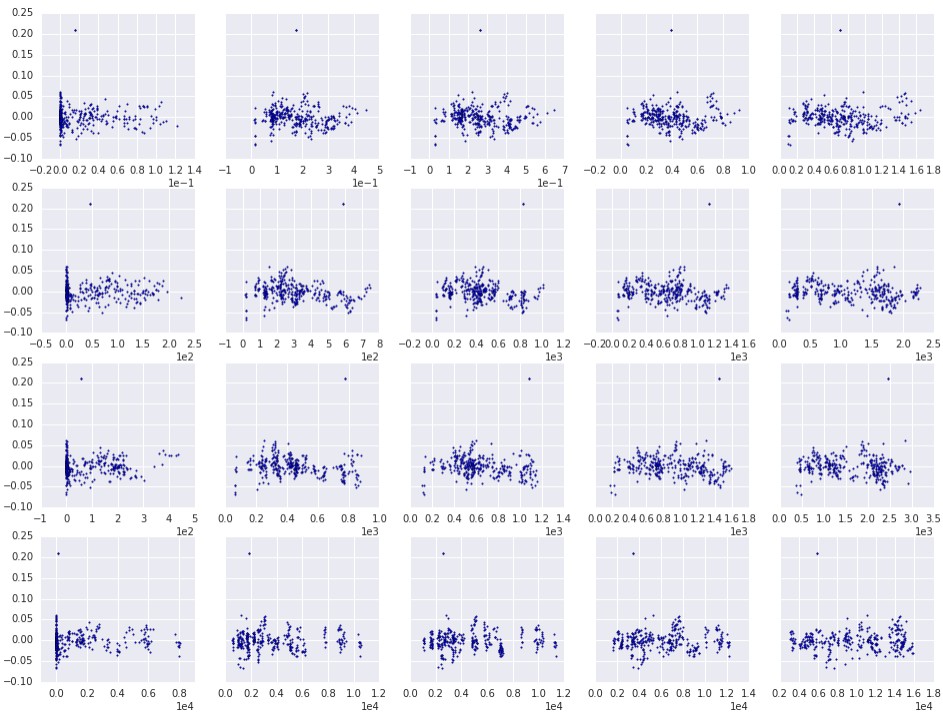

Figure 7: Residual plots for all explanatory variables, row: h0, h1, h2, h3, column: lower fence, $Q_1$, $Q_2$, $Q_3$, upper fence. lower fence is clipped because distance cannot be smaller than 0. The residual is fairly evenly distributed around 0. There is one outlier in this experimental setting as shown in the plots.

|     | lower fence | $Q_1$ | $Q_2$ | $Q_3$ | upper fence |
|-----|-------------|-------|-------|-------|-------------|
| h0  | 80.12       | 8.40  | 59.62 | 141.56 | 248.77     |
| h1  | 65.24       | 109.86 | 343.57 | 700.91 | 1124.43   |
| h2  | 99.06       | 15.47 | 122.36 | 305.88 | 512.69     |
| h3  | 244.07      | 128.45 | 65.58 | 28.10 | 2.34        |

|     | lower fence | $Q_1$ | $Q_2$ | $Q_3$ | upper fence |
|-----|-------------|-------|-------|-------|-------------|
| h0  | 2.85e-17    | 4.00e-3 | 1.46e-13 | 2.65e-27 | 6.32e-42 |
| h1  | 1.34e-14    | 2.60e-22 | 1.04e-52 | 8.12e-83 | 4.55e-107 |
| h2  | 1.59e-20    | 1.03e-4 | 2.53e-24 | 1.29e-48 | 1.42e-68 |
| h3  | 2.40e-41    | 2.78e-25 | 1.16e-14 | 2.13e-7 | 0.127      |

Table 7: F score (top) and p-values (bottom) for all 20 variables. Using $p = 0.05$, the null hypotheses are rejected for every variable except for h3 upper fence.

# 9 APPENDIX: SOME OBSERVATIONS AND CONJECTURES

Everythig here uses the full quartile description.

## 9.1 CROSS ARCHITECTURE COMPARISON

We perform regression analysis with *both* base CNN and ResNet32 on CIFAR-10. The resulting $\bar{R}^2 = 0.91$ and the k-fold $R^2 = 0.88$. This suggests that the same coefficient works generally well across architectures provided they are trained on the same data. Somehow, the distribution at the 3 locations of the networks are comparable even though the depths are vastly different.

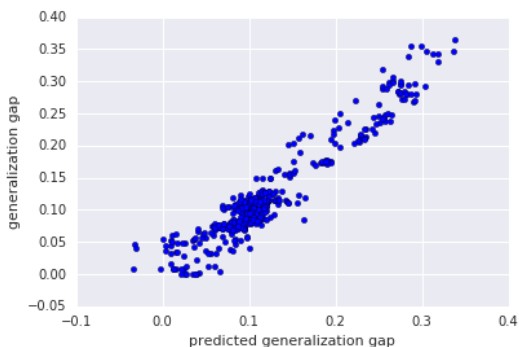

Figure 8: Scatter Plots

## 9.2 CROSS DATASET COMPARISON

We perform regression analysis with ResNet32 on both CIFAR-10 and CIFAR-100. The resulting $\bar{R}^2 = 0.96$ and the k-fold $R^2 = 0.95$. This suggests that the same coefficient works generally well across dataset of the same architecture.

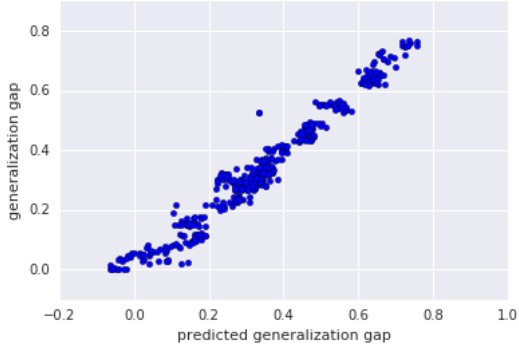

Figure 9: Scatter Plots

## 9.3 CROSS EVERYTHING

We join *all* our experiment data and the resulting The resulting $\bar{R}^2 = 0.93$ and the k-fold $R^2 = 0.93$. It is perhaps surprising that a set of coefficient exists across both datasets and architectures.

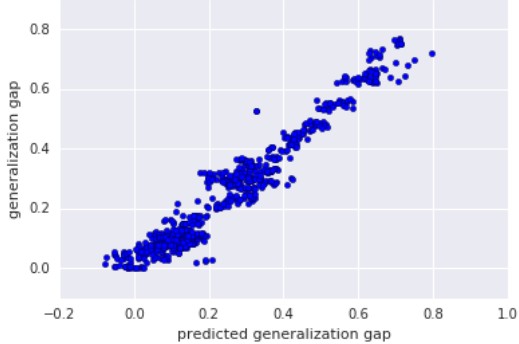

Figure 10: Scatter Plots

## 9.4 IMPLICATIONS ON GENERALIZATION BOUNDS

We believe that the method developed here can be used in complementary with existing generalization bound; more sophisticated engineering of the predictor may be used to actually verify what kind of function the generalization bound should look like up to constant factor or exponents; it may be helpful for developing generalization bound tighter than the existing ones.

