# OpenReview forum: "Predicting the Generalization Gap in Deep Networks with Margin Distributions"
_ICLR.cc/2019/Conference_

### Official Review · AnonReviewer2 · 2018-11-02
**Well written; technically weak**

**Rating:** 6
**Confidence:** 4

**Review:**

After author response, I have increased my score. I'm still not 100% sure about the interpretation the authors provided for the negative distances.

The paper is well written and is mostly clear. (1st line on page 4 has a typo, \bar{x}_k in eq (4) should be \bar{x}^l?)

Novelty: I am not sure whether the paper adds any significant on top of what we know from Bartlett et al., Elsayed et al. since:

(i). The fact that "normalized" margins are strongly correlated with the test set accuracy was shown in Bartlett et al. (figure 1.). A major part of the definition comes from there or from the reference they cite;
(ii). Taylor approximation to compute the margin distribution is in Elsayed et al.;
(iii). I think the four points listed in page 2 (which make the distinction between related work) is misleading: the way I see it is that the authors use the margin distribution in Elsayed et al which simply overcomes some of the obstacles that norm based margins may face. The only novelty here seems to be that the authors use the margin distribution at each layer.

Technical pitfalls: Computing the d_{f,x,i,j} using Equation (3) is missing an absolute value in the numerator as in equation (7) Elsayed et al.. The authors interpret the negative values as misclassification: why is it true? The margin distribution used in Bartlett et al. (below Figure 4 on page 5 in arxiv:1706.08498) uses labeled data and it is obvious in this case to interpreting negative values as misclassification. I don't see how this is true for eq (3) here in this paper. Secondly, why are negative points ignored?? Misclassified points in my opinion are equally important, ignoring the information that a point is misclassified doesn't sound like a great idea. How do the experiments look if we don't ignore them?

Experiments: Good set of experiments. However I find the results to be mildly taking the claims of the authors made in four points listed in page 2 away: Section 4.1, "Empirically, we found constructing this only on four evenly-spaced layers, input, and 3 hidden layers, leads to good predictors.". How can the authors explain this?

By using linear models, authors implicitly assume that the relationship between generalization gaps and signatures are linear (in Eucledian or log spaces). However, from the experiments (table 1), we see that log models always have better results than linear models. Even assuming linear relationship, I think it is informative to also provide other metrics such as MSE, AIC, BIC etc..

---

> ### Author Response · Authors · 2018-11-10
> **Novelty, Experiments, Technical Details**
>
>
> We thank you for your insightful review.
>
> ## NOVELTY ##
>
> R2: “The fact that normalized margins are correlated with generalization was shown in Bartlett Fig 1”.
>
> As you pointed out, both works build on the broad notion of “margin distribution” and “normalization”. However, there are significant differences:
> 1. Margin in Bartlett uses f_i-f_j that can only reflect output margins, as opposed to (f_i-f_j)/||d/dx f_i - d/dx f_j|| that works for any layer.
> 2. We do not use margin distribution itself to predict the generalization gap, but rather distributional features that involve “nonlinear transform” of the distances (quartiles or moments).
> 3. Normalization in Bartlett’s uses norm of weight matrices, which is drastically different from geometric spread of activations (variance) we use (Eqs 4 and 5). Also their cannot be used as-is for residual networks, a drawback not present in our normalization.
>
> These distinctions result in very different predictions of the generalization, as clearly shown in our Fig 2 and Table 1. In fact, the choice of distributional features and normalization are crucial for accurate prediction of the generalization gap.
>
> Finally, we have conducted a far larger scale of experiments, and will be releasing the 700+ realistic models used in the paper so that researchers can easily test generalization theories. This is the first of its kind.
>
>
> ## TECHNICAL ##
>
> # Missing Absolute Value in Eq (3) #
>
> There is no incorrectness; we deliberately adopt “signed distance”. The polarity reflects which side of the decision boundary the point is. Even Eq (7) of Elsayed that you mentioned quickly evolves to signed distance in their Eq (8).
>
> # Why Negative Distance Implies Misclassification #
>
> It was our oversight not to mention that “i” in our Eq (3) corresponds to the ground truth label. We will clarify this in the final version. In this case, f_i-f_j>0 (i.e. distance is positive) implies correct classification and f_i-f_j<0 implies misclassification.
>
> # Why Negative Points are Ignored #
>
> We indeed investigated using negative distances. We observed that:
>
> 1. Modern deep architectures often achieve near perfect classification on training data. Hence, the contribution of negative distances to the full distribution is negligible in most trained models.
>
> 2.  A small fraction of models do have notable misclassification (due to data augmentation or heavy regularization). For these models, we found that margin distribution computed with only positive samples predicted the generalization gap better than (or at par with) the full distribution. However, we observed that the latter is indeed a better predictor of test accuracy (just not the gap). Since we focus our narrative on the generalization gap, we decided to omit these results from the main paper; however, we will include these results in the appendix.
> We also note that there is no technical problem in using margin distribution with only positive samples, e.g. Bartlett’s work “The Sample Complexity of Pattern Classification with Neural Networks” develops a generalization bound by such samples (paragraph above their Theorem 2).
>
>
> ## EXPERIMENTS ##
>
> # Why 4 Layers and Why Even Spacing #
> 1. This leads to a fixed-length signature vector, hence agnostic to the architecture and depth.
> 2. Computing signature across all layers is expensive for large deep models.
> 3. Larger signature would require more pre-trained networks to avoid overfitting in regression phase. Given that each pre-trained network is only one sample in the regression task, creating a large pool of models is prohibitively expensive. Our study with 700 realistic sized pre-trained networks is perhaps already beyond the common practice for such empirical analysis.
> 4. The even spacing is merely a natural choice of minimal commitment and already achieves near perfect prediction (CoD close to 1) is some scenarios. However, it is possible to examine other configurations.
>
> # Log/Linear #
> We are not sure if we understand the question. We provide an answer below, but if this is not what you meant, please let us know. We investigate the use of signature components in two ways: 1. Directly as the input to linear regression, 2. Applying an element-wise log to them before using them as input of the linear regression. In either case, the regression remains linear in optimization variables, but with the log transform we effectively regress the product of signature components to the gap value.
>
> # Other Criteria (MSE, AIC, etc.) #
> We have pointed out that the coefficient of determination already captures the MSE along with the scale of the error; however, for completeness, we will include this result in the appendix. We report k-fold cross validation results as well, which is known to be asymptotically equivalent to AIC (Stone M. (1977) An asymptotic equivalence of choice of model by cross-validation and Akaike’s criterion)

---

### Official Review · AnonReviewer1 · 2018-11-03
**A nice empirical paper with good intuitions and encouraging results**

**Rating:** 9
**Confidence:** 4

**Review:**

This paper does not even try to propose yet another "vacuous" generalization bounds, but instead empirically convincingly shows an interesting connection between the proposed margin statistics and the generalization gap, which could well be used to provide some "prescriptive" insights (per Sanjeev Arora) towards understanding generalization in deep neural nets.

I have no major complaints but for a few questions regarding clarifications,
1. From Eq.(5), such distances are defined for only one out of the many possible pairs of labels. So when forming the so-called "margin signature", how exactly do you compose it from all such pair-wise distances? Do you pool all the distances together before computing the statistics, or do you aggregate individual statistics from pair-wise distances? And how do you select which pairs to include or exclude? Are you assuming "i" is always the ground-truth label class for $x_k$ here?

2. In Eq.(3), the way you define the distance (that flipping i and j would change the sign of the distance) is implying that {i, j} should not be viewed as an unordered pair, in which case a better notation might be (i, j) (i.e. replacing sets "{}" with tuples "()" to signal that order matters).

And why do you "only consider distances with positive sign"? I can understand doing this for when neither i nor j corresponds to the ground-truth label of x, because you really can't tell which score should be higher. But when i happens to be the ground-truth label, wouldn't a positive distance and a negative distance be meaningful different and therefore it should only be beneficial to include both of them in the margin samples?

And a minor typo: In Eq.(4), $\bar{x}_k$ should have been $\bar{x}^l$?

---

> ### Author Response · Authors · 2018-11-10
> **Addressing your comments**
>
> We would like to thank you for your review and suggestions. We are very glad that you liked the empirical analysis of generalization gap and margin distribution statistics. On that note, while not mentioned in the paper, we are in preparation to release the  700+ models we used in the paper as a dataset where researchers can easily test theories on generalization. We believe this will be one of the first datasets for studying generalization on realistic and modern network architectures and we hope it will be instrumental in the ongoing generalization research.
>
>
> ## Construction of Signature from Pairwise Distances (i,j) in Eq (5) ##
>
> For computational efficiency, we picked we pick ground truth label as "i" (as you correctly pointed out), and the highest non-ground truth logit as "j", and compute the distance between the two classes. While aggregating all pairwise distance might be more comprehensive, the complexity scales roughly quadratically with the number of classes. As such, we made the design choice to use the top two classes. In cases where the class with the highest logit is not the ground truth (hence misclassification with negative distance), we discard the data point. We will further explain this choice below. We mention this detail in the text but we will make sure it is more clear.
>
>
> ## Notation (i,j) instead of {i,j} to Emphasize Orderedness ##
>
> Thank you for the suggestion. We agree and will incorporate this in the revision to avoid confusion.
>
>
> ## Why Only Positive Distances in Margin Distribution ##
>
> You are right that when “i” is the ground truth label, the sign of the distance indicates whether the point is correctly classifier or is misclassified.
>
> We indeed investigated using negative distances when computing the margin distribution. We observed that:
>
> 1. Modern deep architectures often achieve near perfect classification on training data. Hence, the contribution of negative distances to the full distribution is negligible in most trained models.
>
> 2.  A small fraction of models do have notable misclassification (due to data augmentation or heavy regularization). For these models, we found that margin distribution computed with only positive samples predicted the generalization gap better than (or at par with) the full distribution. However, we observed that the latter is indeed a better predictor of test accuracy (just not the gap). Since we focus our narrative on the generalization gap, we decided to omit these results from the main paper; however, we will include these results in the appendix.
> We also note that there is no technical problem in using margin distribution with only positive samples, e.g. Bartlett’s work “The Sample Complexity of Pattern Classification with Neural Networks” develops a generalization bound by such samples (paragraph above their Theorem 2).
>
>
> ## Typo ##
>
> Thank you for pointing out the typo. It will be fixed in revision.

---

### Official Review · AnonReviewer4 · 2018-11-11
**An empirical study towards the prediction power based on the margin distribution at each layer.**

**Rating:** 5
**Confidence:** 4

**Review:**

The author(s) suggest using geometric margin and layer-wise margin distribution in [Elsayed et al. 2018] for predicting generalization gap.

pros,
a). The author shows large experiments to support their argument.

cons,
a). No theoretical verification (nor convincing intuition) is provided, especially for the following questions,
    i) what benefit can be acquired when using geometric margin defined in the paper.
    ii) why does normalization make sense beyond the simple scaling-free reason. For example, spectral complexity as a normalization factor in [Bartlett et al. 2017] is proposed from the fact, that the Lipschitz constant determines the complexity of network space.
    iii) why does the middle layer margin can help?
    iv) why a linear (linear log) relation between the statistic and generalization gap.

Further question towards experiment,
i) I don't think your comparison with Bartlett's work is fair. Their bounds suggest the gap is approximately Prob(0<X<\gamma) + Const/\gamma for a chosen \gamma, where X is the normalized margin distribution. I think using the extracted signature from margin distribution and a linear predictor don't make sense here.
ii) If you do regression analysis on a five layers cnn, can you have a good prediction on a nine layers cnn (or even residue cnn)?

Finally, I'm not sure the novelty is strong enough since the margin definition comes from [Elsayed et al. 2018] and the strong linear relationship has been shown in [Bartlett et al. 2017, Liao et al. 2018] though in different settings.

---

> ### Author Response · Authors · 2018-11-27
> **Addressing your comments.**
>
> Thank you for the review. We address your concerns below.
>
> #What benefit can be acquired when using geometric margin defined in the paper.#
> The geometric distance is the actual distance between a point “x” and the decision boundary f(x)=0, i.e. d1=min_x ||x|| s.t. f(x)=0.This term is usually used in contrast to functional distance defined as d2=f(x). If x is on the decision boundary, d1=d2=0, but otherwise d1 and d2 can differ. Note that d2 can change by simple reparametrization. For instance, consider a linear decision boundary f(x)=w.x. In this case, geometric distance d1=f(x)/||w|| and d2=f(x). Let F(x)=(c*w).x, i.e. just scaling the weights by factor c. This does not change the decision boundary. For such F, d1 remains the same, but d2 scales with c. One can force a condition to make margins equal in both scenarios: by making the closet point to the decision boundary to have distance 1. However, this requires introducing an inequality per point, similar to SVMs. With geometric margin, we can work with an unconstrained optimization and directly apply gradient descent or SGD.
>
> #Why does normalization make sense?#
> Our normalization allows direct analysis of the margins across different models with the same topology (or different datasets trained on the same network), which is otherwise difficult due to the positive homogeneity of ReLU networks. For example, suppose we have two networks with exactly the same weight, and then in one of the networks, we scale weight_i by constant positive factor c and the weight_{i+1} by 1/c (i is a layer index), the predictions of the two networks remain the same; however, their unnormalized margin distribution will be vastly different and the normalized version will be exactly the same.
>
> #Why does the middle layer margin can help? #
> There is no reason we can assume a-priori that maximizing only input or output margin (for example) is enough for good generalization. As shown in our ablation results in Tables 1 and 4, the combination of multiple layers performs significantly better. If we cut a deep network at any stage, we can treat the first half of the network as a feature extractor and the second half as the classifier. From this perspective, the margins at middle layer can be just as important as the margins in the output layer or input layer. Lastly, we note that Elsayed et. al. show that optimizing margin at multiple layers provides significant benefits for generalization and adversarial robustness.
>
> #Why a linear (linear log) relation between the statistic and generalization gap.#
> We are not claiming this is the true relationship between the statistics and the generalization gap. The true relationship may very well be nonlinear and one could perform a nonlinear regression to predict the gap, but it would need regularization and more data to avoid overfitting while a linear combination of simple distributional features already attains high quality prediction (according to CoD, k-fold cross validation and MSE) across 700+ pretrained models. This suggests that a linear relationship is indeed a very close *approximation*.
>
> #I don't think your comparison with Bartlett's work is fair. Their bounds suggest the gap is approximately Prob(0<X<\gamma) + Const/\gamma for a chosen \gamma, where X is the normalized margin distribution. I think using the extracted signature from margin distribution and a linear predictor don't make sense here.#
> We assume the reviewer is referring to theorem 1.1 of Bartlett et al. If one wishes to compute the gap to be the inside of the soft big O, the result will be much larger than the error emitted by our prediction, and will require picking appropriate gamma and delta values. We further note the following: the case study of Bartlett et. al. (section 2) explicitly show in their diagrams (Figures 2 and 3) the normalized distribution as evidence of generalization prediction power (instead of the bound itself) and this normalized distribution is closely related to but is not directly their bounds (they drop the log terms); extracting the statistics in a sense quantifies their case study. Before submitting the paper, we also had personal communication with one of the authors of Bartlett et. al., and the author agreed that our comparison was fair.

---

> > ### Author Response · Authors · 2018-11-27
> > **Addressing your comments (contd.)**
> >
> > #If you do a regression analysis on a five layers cnn, can you have a good prediction on a nine layers cnn (or even residue cnn)#
> > In the Appendix (Section 9.1 and 9.2), we already show both cross-architecture and cross-dataset comparisons, which achieve good predictive accuracy but worse than the result on a single architecture. However, when we tried using the result from cnn alone to predict the generalization gap of residual network or vice versa (not included in the paper), the result does not signify any interesting correlation. Nevertheless, we would like to emphasize that the regression is shared (and gives an accurate prediction) across other significant changes such as channel sizes, batchnorm/group norm, regularization, learning rate, dropout change (presented in appendix section 6)
> >
> > # Novelty #
> > As you correctly pointed out, our work and Barlett et. al. build on the broad notion of “margin distribution” and “normalization”. However, there are significant differences:
> > 1. Bartlet’s definition of margin relies only on f_i-f_j, which only reflects margin in the output space, as opposed to (f_i-f_j)/||d/dx f_i - d/dx f_j|| which approximates margin in input (or any hidden) space.
> > 2. The normalization used in Bartlett et al. is a complexity measure which is drastically different from our normalization that captures more direct geometric properties of the activations. Specifically, Bartlett’s normalization relies on the spectral complexity of a network which involves spectral norm of weight matrices and reference matrices. In our work, the normalization is defined based on the total variance of the activations of the hidden layers directly (Eqs 4 and 5).
> > 4. Barlett et. al. do *not* show any linear relationship between margin and test performance or gap.
> > The above distinctions lead to very different predictions on the generalization gap as shown in our results (Figure 2 and Table 1). In fact, the choice of distributional features and normalization scheme are crucial for accurate prediction of the generalization gap.
> >
> > Furthermore, we note again that the normalization scheme of Bartlett et. al. cannot be used as-is for residual networks and is not applicable to hidden layers, a drawback not present in our normalization. Finally, we have conducted a far larger scale of experiments as compared to Bartlett et. al. to verify the effect of each prediction scheme of the generalization gap. Like we mentioned in our response to reviewer 1, we will be releasing the 700+ realistic models we used in the paper as a dataset where researchers can easily test theories on generalization, which is one of the first of its kind.
> >
> > Regarding Liao et. al. 2018, as stated in the paper, their proposed normalized loss leads to a significant *decrease* in output margin confidence, which is the opposite of what is desirable. Furthermore, normalized cross-entropy loss is different from margin-based loss, so we do not think their observation takes away the novelty of our paper just because both works illustrate linearity.

---

### Public Comment · (anonymous) · 2018-11-01
**Some comments**

Introducing the theory of margin distribution into the framework of deep learning is an interesting idea. And it seems that there is a related work [Optimal margin Distribution Network, Submission to ICLR 2019], which has tried to design a new loss function based on margin distribution and theoretically proved its generalization effect. As I know, the influence of margin distribution has always been a concern for generalization theory. [Schapire, 1998] [Wang, 2011] [Gao, 2013], and there are several new algorithms based on the theory of margin distribution in both SVM [Zhang, 2017] and Clustering [Zhang, 2018] frameworks. I think that authors should read these papers and add references to them.
Regarding the content of the paper, I am confused about the linear (or log() ) estimation of the generalization gap: "$\hat{g} = a^T \phi(\theta) + b$". Does this formula have a theoretical analysis or some statistical models to explain it? It seems unreasonable to directly explain the relationship between margin distribution and generalization with a simple linear relationship. I expect that the authors can theoretically give a formula to explain the relationship between the generalization gap and the margin distribution.


[Optimal margin Distribution Network, Submission to ICLR 2019] Anonymous. “Optimal margin Distribution Network” Submitted to International Conference on Learning Representations 2019
[Schapire, 1998] Schapire, R., Freund, Y., Bartlett, P. L., Lee, W. Boosting the margin: A new explanation for the effectives of voting methods. Annuals of Statistics 26 (5), 1651–1686. 1998
[Wang, 2011] Wang, L. W., Sugiyama, M., Yang, C., Zhou, Z.-H., Feng, J. “A refined margin analysis for boosting algorithms via equilibrium margin.” Journal of Machine Learning Research 12, 1835–1863. 2011
[Gao, 2013] Gao, W., and Zhou, Z.-H. "On the doubt about margin explanation of boosting." Artificial Intelligence 203, 1-18. 2013
[Zhang, 2017] Zhang, T., Zhou, Z.-H. "Multi-Class Optimal Margin Distribution Machine." International Conference on Machine Learning. 2017.
[Zhang, 2018] Zhang, T., Zhou, Z.-H. "Optimal Margin Distribution Clustering." Proceedings of the National Conference on Artificial Intelligence, 2018.

---

> ### Author Response · Authors · 2018-11-01
> **Thanks for comments**
>
> Thank you for your helpful comments.
>
> ### References ###
>
> 1. We agree that the interaction of margin and generalization has been subject to a great amount of research in classical ML literature. This makes it impossible to provide a comprehensive survey in a conference paper. So we had to narrow the scope of related works to recent papers that address generalization/margin in the case of *deep* models. Nonetheless, we will be happy to include the references on SVM and clustering that you suggested.
>
> 2. Regarding the other ICLR2019 submission you mentioned, obviously we were not aware of it prior to ICLR submission deadline (and it is not available on arxiv either). We are aware of that submission, but it seems to have some issues (reading the comments for the paper).
>
> ### Linear Assumption ###
>
> 1. Regarding your suspicion of linear relationship between margin and generalization gap: we are not directly relating the two using a linear map. Note that we are converting the margin distribution to a feature vector via a nonlinear map (quartiles/moments), and it is these features that are regressed to the generalization gap by a linear map. This is a widely used idea for nonlinear regression; e.g. as in kernel SVM for regression (nonlinear feature space followed by linear fitting). One could also train a nonlinear (deep) neural net to predict the gap, but it would need regularization and more data to avoid overfitting while a linear combination of simple distributional features already attains high quality prediction (see next point) across ~700 pretrained models. The latter suggests that a linear relationship is indeed a very close approximation.
>
> 2. The point of the paper is not to claim an optimal feature set, but to leverage *simple* and *easy to compute* features that could be extracted from the distribution (like quartiles or moments) can yet give a reasonable prediction of the generalization gap that is much better than recent theoretical upper bounds in the literature. We hope this could be a step toward constructing *practical* algorithms for improving generalization in deep networks. Regarding mathematical proof for why these features should explain the generalization gap: while such results would be very interesting, it is quite ambitious if not impossible. Nevertheless, we assess the quality of the linear fit using one of the standard statistical tools created for this purpose: Coefficient of Determination (CoD).  As mentioned in the paper, in some scenarios we observe CoD=0.97 (max is 1.0) which indicates a reasonably good fit.

---

### Author Response · Authors · 2018-11-27
**Summary of revisions and responses to all reviewers.**

We thank all the reviewers for their comments, suggestions and questions. We have responded to each reviewer’s individual comments below. We have modified the paper as follows to address common questions posed by the reviewers:

1. Using negative examples: we have added linear fits to both test accuracy and generalization gap and shown comparisons with and without negative examples. Table 3 in Appendix 7 (page 13) shows these results. We see that using negative margins predicts accuracy better than the generalization gap. However, as noted above, we chose to predict generalization gap, and in that case, a log relationship provides much stronger prediction, but log transform cannot use negative margin values.
2. To answer R2’s question about the importance of hidden layers, we show in Table 4, Appendix 7, the results of fitting every single layer and compare to fitting all layers together. No single layer, input, hidden or output performs as well as the combination. We also provide intuition for why it is important from a theoretical perspective to use margins at hidden layers (Section 3).

We have added to the main body or appendix of the paper a few smaller edits:
1. typos identified by R1 (Eq. 4)
2. more compact notations for Table 1

Clarifying explanations:
1. Why we choose to discard negative margins (Sec. 3.1)
2. Why we use both a linear and log regression model (Sec. 3.3)
3. Mean square error computations (Tables 1, 3, and 4)
4. Why we chose evenly spaced layers for our margin computations. (end of Section 3.2)
5. Added references suggested by reviewers and commenter.

Lastly, we will release all the trained CIFAR-10 and CIFAR-100 models. We hope this work along with the model dataset will open up interesting avenues for future research.

We hope the rebuttal and revision have addressed the reviewers’ questions and comments.

Thank you!

---

### Meta-Review · Area_Chair1 · 2018-12-16
**A layer-wise geometric margin distribution is used to calibrate the generalization ability, with extensive experimental support yet lacking a theory.**

**Confidence:** 4
**Recommendation:** Accept (Poster)

**Metareview:**

The paper suggests a new measurement of layer-wise margin distributions for generalization ability. Extensive experiments are conducted. Though there lacks a solid theory to explain the phenomenon. The majority of reviewers suggest acceptance (9,6,5). Therefore, it is proposed as probable accept.